# Is There a Role for Cytoreductive Surgery and Hyperthermic Intraperitoneal Chemotherapy in Peritoneal Carcinomatosis Due to Uterine Cancer?

**DOI:** 10.3390/jpm12111790

**Published:** 2022-10-30

**Authors:** Özgül Düzgün, Murat Kalin

**Affiliations:** Department of Surgical Oncology, Istanbul Ümraniye Training and Research Hospital, Health Sciences University, 34766 İstanbul, Turkey

**Keywords:** uterine cancer, peritoneal carcinomatosis, cytoreductive surgery, hyperthermic intraperitoneal chemotherapy

## Abstract

Background: Uterine canceris one of the most common pelvic tumors in females. Advanced stage uterine cancer only represents 15% of newly diagnosed cases; however, they are related with poor prognosis. Our aim was to analyze the benefits of cytoreductive surgery and hyperthermic intraperitoneal chemotherapy in peritoneal carcinomatosis due to uterine cancer. Methods: At the Istanbul Umraniye Training and Research Hospital, Surgical Oncology Clinic, morbidity, overall survival and survival without progression were analyzed over the 5-year follow up. Twenty-two cases who had undergone cytoreductive surgery and hyperthermic intraperitoneal chemotherapy due to uterine-peritoneal carcinomatosis were included in this study. Cases were followed up in terms of postoperative morbidity-mortality, disease-free survival and overall survival. The cut off value for the peritoneal carcinomatosis index score was set at 15. Intraperitoneal chemotherapy consisting of cisplatin and doxorubicin was applied to all patients for 60 min after the suturation of the abdomen. Results: Median age of the patients was 64.6 (43–72). Average PCI score was 12.8 (3–15). CC score was 0 in 16 (72.7%) cases, 1 in 3 cases and 2 in 3 cases. Of these patients, 12 of them were previously operated upon. Median stay at the hospital was 13.1 days. No major complications due to chemotherapy were reported. A Clavien–Dindo Grade 3 complication was observed in seven (31.8%) patients. Mortality was not observed in patients during their stay at the hospital. The 5-year disease-free survival and overall survival rates were 36.8 (36%) months and 45.3 (57%) months, respectively. Conclusions: We think that due to longer disease-free survival and overall survival, cytoreductive surgery and hyperthermic intraperitoneal chemotherapy should be preferred in peritoneal carcinomatosis due to uterine cancer patients having low peritoneal carcinomatosis index scores and manageable complication rates. However, prospective randomizedtrials with a high number of cases are needed for this subject.

## 1. Introduction

Uterine cancer (UC) is one of the most common pelvic tumors in females. The lifelong risk for UC is around 4% in females [1]. It is usually seen in women around the age of 60–70. UC has a good prognosis following the combination treatment of surgery, chemotherapy, radiotherapy and hormonal treatment; and only 2% of cancer related deaths are due to UC [2]. Advanced stage UC only represents 15% of newly diagnosed cases; however, they are related to poor prognosis. The 5-year-survival rates of females with local (49–66%) or distant (20–25%)metastasis to the peritoneum decrease, resulting in median survival of less than 1 year [3,4].

Cytoreductive surgery (CRS) and hyperthermic intraperitoneal chemotherapy (HIPEC), having a place in the guidelines, can be successfully applied to cases with pseudomyxoma peritonei originating from the appendix without extra-abdominal metastasis, and in selected patients with ovary and colon cancers having low peritoneal carcinomatosis index (PCI)scores. In the literature, there are no current Randomized Controlled Trials RCTs regarding the efficacy of CRS and HIPEC in UC. Retrospective series from experienced centers provide an insight for the treatment of synchronous and metachronous peritoneal metastasis (PM) to the peritoneum due to UC with CRS and HIPEC; however, the low number of cases involved limits the credibility of the statistics [5,6,7,8].

Regarding these limitations, we aimed to analyze the 5-year results of our patients with peritoneal carcinomatosis (PC) due to UC in our tertiary center. Our aim was to analyze the benefits of SRC and HIPEC in PC due to UC. Morbidity, overall survival and survival without progression were analyzed during the 5-year follow up.

## 2. Materials and Methods

Between May 2017 and May 2022, at the Health Sciences University İstanbul, Ümraniye Training and Research Hospital Surgical Oncology Clinic, the data of 360 patients who had undergone CRS and HIPEC surgery because of intra-abdominal metastases were collected prospectively and evaluated retrospectively. Written informed consent was obtained from the study participants and the ethics committee of the Health Sciences University İstanbul, Ümraniye Training and Research Hospital approved the study (numbered 2022/192). A total of 30 patients were diagnosed with UC; however, 8 of the patients had a PCI score higher than 15 and therefore these patients were excluded from the study. There were 22 cases with PC due to UC recorded in the surgical oncology clinic system. Twenty-two cases who had undergone CRS and HIPEC due to uterine PC were included in this study. Demographic data of the cases, such as age, comorbidities, American Society of Anesthesiologists (ASA) score, Eastern Cooperative Oncology Group score, body surface area (BSA), previous CRT story, duration of surgery, PCI, completeness of cytoreduction (CC) score, fluid resuscitation perioperatively, the need of erythrocyte suspension and fresh frozen plasma, amount of urine, and duration of intensive care unit (ICU) and hospital stay were evaluated in terms of postoperative morbidity–mortality, disease-free survival (DFS) and overall survival (OS). All patients were operated with the approval of the multidisciplinary tumor council.

Abdomen and pelvic MRI, oncologic PET-CT and tumor markers (AFP, CEA, CA 19-9, CA-125) were evaluated preoperatively. Diagnostic laparoscopy was used preoperatively in order to calculate the PCI score. Cut off value for PCI score was set as 15. Operations were continued in patients with PCI score 15 or lower. Patients with PCI score higher than 15 were directed to neoadjuvant chemotherapy and re-evaluated after neoadjuvant chemotherapy and operated upon afterwards.

Various prognostic scoring systems are needed for the selection of patients for this highly invasive surgery. PCI is the most commonly used one today. The lower the score, the higher the survival. The main purpose here was to provideR0 resection, which is not to leave a tumor behind macroscopically. Contraindications for CRS and HIPEC include extra-abdominal metastases, low Karnofsky performance scores, severe cardiac, pulmonary, hepatic or renal dysfunctions. In addition, extensive small bowel, mesenteric involvement, multiple liver metastases and para-aortic lymph node involvement are also considered as a contraindicationas they do not contribute to the survey.

During the operation, the modified lithotomy position was used. The operation was started laparoscopically and PCI score was calculated. Afterwards, midline incision was made starting from the xiphoid process to the pubis. After the incision, PCI score was calculated. All the tumoral masses in pelvic area and in other sites of the abdomen were excised. CRS was performed as previously described by Sugarbaker [9]. Anastomoses were made before HIPEC. In addition, anastomoses ofileostomy, colostomy were made before HIPEC.

HIPEC surgical drainage was placed in the bilateral subdiaphragmatic areas, epigastric area, and pelvic areas. The abdomen was sutured after the placement of heat probes in the pelvic and epigastric areas. Immediately after the abdominal closure, cisplatin (75 mg/m^2^BSA) + doxorubicin (15 mg/m^2^BSA) in 0.9% NaCl solution was injected intra-abdominally and intraperitoneally in 43° and 1200 cc/h turns for 60 min. HIPEC was performed as a closed technique in all cases. During this procedure, intra-abdominal body temperature was measured using a probe placed in the esophagus by the Belmont Hyperthermia Pump (Belmont Instrument Corporation, Billerica, MA, USA). After the procedure, the patients were transferred to the ICU.

### Statistical Analyses

The data obtained from raw data were recorded to IBM SPSS Statistics 22.0 (IBM SPSS, Turkey) software and analyzed. The numerical data obtained were summarized in tables as arithmetic mean ± standard deviation, minimum, maximum, and range values. The nominal and ordinal data were evaluated as frequency and percentages. Finally, overall survival was calculated by using Kaplan–Meier survival analysis available in the same software.

## 3. Results

CRS and HIPEC were performed on 22 patients with PC due to UC. CRS was re-done and HIPEC was performed on three (13.6%) of these patients due to fact that these patients had recurrent metastatic disease. Earliest recurrence was observed in the postoperative 12th month. Of these patients, 12 of them were previously operated on (total abdominal hysterectomy (TAH)). The median age of the patients was 64.6 (43–72), the median ASA score was 1.7 (1–3),median body surface area (BSA) was 180.4 (142–199) and median Karnofsky performance score was 82 (70–100). Of these patients, 14 (63.6%) of them had preoperative chemotherapy. The median stay at the hospital was 13.1 days (5–49), Table 1.

The average operation time was 5.6 h (3–8), average PCI score was 12.8 (3–15). CCscore was 0 in 16 (72.7%) cases, 1 in 3 cases and 2 in 3 cases. Total small intestine and colon anastomosis numbers were 1 in 7 (31.8%)cases and 2 in 5 (22.7%) cases. An average of 3400 cc (2000–5500) crystalloids, 650 cc (500–1000) colloids, 1.1 units (0–4) of erythrocyte suspension and0.6 units of fresh frozen plasma was transfused to patients perioperatively. Average urine output was 740 cc (280–2100) and average blood loss was 590 cc (200–2400), Table 2.

Macroscopic tumor was not left behind after the complete organ resections were completed in CRS with the exception of two patients, Table 3. Intraperitoneal chemotherapy consisting of cisplatin and doxorubicin was applied to all patients for 60 min after the suturation of the abdomen. No major complication due to chemotherapy was reported.

The Clavien–Dindo (CD) scoring system was used in order to classify the complications. During the postoperative period, a CD grade 3 complication was observed in seven (31.8%) patients. A colorectal anastomosis leak (CD grade 3b) was observed in two patients and these were reoperated and a Hartmann colostomy was performed. One case was reoperated due to bleeding (CD grade 3b). One case had urine leakage from the bladder wall (CD grade 3a)and was treated with bilateral nephrostomy. One case of evisceration (CD grade 3a)was observed and the abdomen was sutured under local anesthesia. Pleural effusion (CD grade 3a)was observed in two cases and was treated by placement of drainage catheters by interventional radiology. No CD grade 4 complication was observed. Mortality was not observed in patients during their stay at the hospital, Table 4. In the pathology reports, seven (31.8%) cases were reported as endometrial carcinoma, five cases(22.7%) were reported as carcinosarcoma, five cases(22.7%) were reported as leiomyosarcoma, three (13.6%) cases were reported as endometrial stromal sarcoma, one case (4.5%)was reported as undifferentiated sarcoma and, lastly, one case (4.5%) was reported as serous carcinoma. All cases were followed postoperatively after their medical and radiological treatment was finished, Table 5. A locoregional relapse was observed in three cases; these relapses were observed at postoperative 8th, 10th and 12th months, respectively, and these cases were reoperated. One case was considered as exitus due to COVID-19 within 30 days postoperatively. The median follow up was 32 (6–60) months. The 5 year disease-free survival and overall survival rates were 36.8 (36%) months and 45.3 (57%) months, respectively, Table 6.

## 4. Discussion

Surgery is regarded as a cornerstone during the treatment of UC. The main goal of the surgery is to treat the primary cancer (usually involving hysterectomy and bilateral salpingo-oophorectomy) and evaluate the surgical stage regarding the excision of lymph nodes in order to guide the adjuvant chemo-radiotherapy [10,11]. Two out of three patients operated on due to UC are referred to hospital again in an average of 2 years, due to recurrence of the disease [3]. The majority of these patients are diagnosed as PC and the classical approaches for these patients are chemotherapy, chemoradiotherapy, salvage radiotherapy or targeted therapy [12].

In the last two decades, CRS and HIPEC have a proven efficacy in mesothelioma and pseudomyxoma peritonei along with colon and ovary tumors and also PC patients [13]. Van driel et al. found, in a multicentric phase 3 randomized controlled trial in patients with Stage III epithelial ovarian cancer, that adding HIPEC to intermittent cytoreductive surgery resulted in longer relapse-free and overall survival and no higher rates of adverse events than surgery alone [14]. PC due to UC also has a place among these treatments. There is no common consensus upon the usage of CRS + HIPEC in PC due to ovary cancers and it is a newly used treatment regimen; due to the fact that the uterus is located in the pelvic area, the diagnosis and treatment of local recurrences and PC can be difficult. These difficulties can be overcome by radiologists experienced’ in MRIs, PET-CT and diagnostic laparoscopy. In our study, we wanted to emphasize the fact that CRS and HIPEC could be useful in PC arising from UC. The main goal here was to detect the patient group who could benefit from CRS and HIPEC. Patients with low PCI score, an age younger than 70, and Karnofsky performance score higher than 70 can benefit from this procedure. Although performed in a limited series, CRS and HIPEC was used in patients with PC due to UC. Tempfer and al. detected 68 patients in the literature through their literature research of eight studies. In this patient population, 70% of patients had CC-0 resection and reported the DFS and OS to be 7–18 months and 12–33 months, respectively. They concluded that CRS and HIPEC is safe in patients with PC due to UC [1].

Navarro Barios et al. performed CC-0 resection on 41 of 43 patients and reported the 5-year DFS and OS as 23% and 34%, respectively [15]. Cornali et al. performed CC-0 resection on 22 of 33 patients and reported a medianOS to be 33 months [16]. In the research by Gomes David et al., they compared two groups: one group consisted of 44 patients in which CRS and HIPEC were performed and the other group consisted of 90 patients in which only CRS was performed. They concluded that there was no significant difference regarding the DFS and OS between the two groups [17]. We performed CRS and HIPEC on 22 patients and 16 of them (72%) were CC-0 resection. The DFS and OS of the 5-year results were 36% and 57%, respectively. Our CC-0 resection score was concomitant with the literature and concluded the DFS and OS to be a little higher than the literature. We concluded that these results were such due to the fact that we only operated on patients with PCI scores less than 15 and diagnostic laparoscopy was used perioperatively in every patient. Furthermore, in our clinic, all of the patients were operated on by the same surgeon.

Uterus corpus sarcoma is a rarely seen mesenchymal tumor. It constitutes 7% of all EC. Uterine sarcomas respond poorly to systemic chemotherapy and they are characterized with high recurrence levels. Although the usage of CRS and HIPEC is controversial in peritoneal sarcomatosis, it is accepted as a major treatment modality in many centers around the world [18,19]. Diaz Montes et al. applied CRS and HIPEC to 7 cases of a26 case series in their retrospective single-centered study over 11 years. They reported the superiority of43 months of survival with CRS and HIPEC treatment in comparison to 35 months of survival with conventional therapy [20]. In our series, 13 (59%) of our cases consisted of uterine sarcomas.

Commonly used therapeutics in intraperitoneal sarcomas and gynecologic-originated tumors are cisplatin or a combination of cisplatin with doxorubicin. The dosage changes according to 50–75 mg/body surface area M2, the duration varies from 60 to 90 min and the temperature is between 41 and 43 degrees, changing from center to center. Due to these variations, there is no consensus upon the HIPEC dosage, duration and chemotherapeutic type [1]. In our study, we used cisplatin 75 mg/BSA + doxorubicin 15 mg/BSA intraperitoneally over 60 min, concomitant with the literature.

One of the main problems with CRS and HIPEC treatment in PC patients due to intra-abdominal tumors is high morbidity rates. In the last decade, these morbidity rates lowered from 50 to 20%. Mortality rates need to be lowered to 5%. In their retrospective multicentric series, Gomes David et al. reported a complication rate of20% which were CD grade3 or higher. They did not report mortality [17]. In their 33 case series, Cornali et al. reported CD grade 3 or more complication rates of 3% and a mortality rate of 0%, whereas Delotte et al. did not report a CD grade 3 or more complication rate [7,16]. In our center, seven (31%) cases had a grade 3 or more complication and one (4.5%) of our cases was considered exitus due to COVİD 19. Our complication rate was higher compared to that in the literature and we think that this is due to intestinal and organ resections being performed.

There are several limitations to this article. The study was retrospective; the number of cases was small; there was no a control group with which to compare; pathologies were not chosen homogeneously, thus increasing the possibility of bias. Furthermore, previous surgery and chemotherapy history and different PCI scores made it hard to differentiate between the local effect of HIPEC and progression-free survival rate.

## 5. Conclusions

We think that due to the longer DFS and OS, CRS and HIPEC should be preferred in PC due to UC patients having low PCI scores and manageable complication rates. However, prospective randomized trials with a high number of cases are needed for this subject.

## Figures and Tables

**Table 1 jpm-12-01790-t001:** Demographic data of patients.

Age	64.6 ± 10.26
ASA score	
1	6 (27.2%)
2	6 (27.2%)
3	10 (45.4%)
Karnofskyscore	82 ± 10.34
BSA	180.4 ± 12.28
Preoperative chemotherapy	14(63.6%)
Hospitalization day	13.1 (5–49)
Data are expressed as mean ± SD and *n* (%).	

Abb: ASA: American Society of Anesthesiologists, BSA: body surface area.

**Table 2 jpm-12-01790-t002:** Perioperative findings.

Operation Time (hour)	5.6 (3–8)
PCI score	12.8 (3–15)
Residual tumor CC	
CC-0 (no residual nodules)	16 (72.7%)
CC-1 (residual < 2.5 mm)	3 (13.6%)
CC-2 (residual > 2.5 mm)	3 (13.6%)
Number of anastomosis	
0	10 (45.4%)
1	7 (31.8%)
2	5 (22.7%)
Perioperative fluid	
Crystalloid	3400 (2000–5500)
Colloid	650 (500–1000)
Erythrocyte suspension	1.1 (0–4)
Fresh frozen plasma	0.6 (0–4)
Urine	740 (280–2100)
Hemorrhage	590 (200–2400)

Abb: PCI: peritoneal carcinomatosis index, CC: completeness of cytoreduction.

**Table 3 jpm-12-01790-t003:** Organ resections.

TAH + BSO	10 (45.4%)
Partial vaginectomy	1 (4.5%)
Partial bladder excision and primary closure	5 (22.7%)
Partial ureter resection and ureteroneocystostomy	3 (13.6%)
Total cystectomy + ileal conduit	1 (4.5%)
Colon resection	9 (40.9%)
Rectum resection	8 (36.3%)
Small bowel resection	9 (40.9%)
Peritonectomy	22(100%)
Splenectomy	2(9.1%)
Cholecystectomy	3 (13.6%)
Partial pubic bone excision	1 (4.5%)
Stoma status	
End ileostomy	1 (4.5%)
End colostomy	2 (9.1%)

Abb: TAH + BSO total abdominal hysterectomy + bilateral salpingo oopherectomy.

**Table 4 jpm-12-01790-t004:** Grade 3Clavien–Dindo (CD) complications classification and management.

Complication	Cases	Management
Leakage in colorectal anastomosis (CD grade 3b)	2	Hartmann end colostomy
Postoperative hemorrhage (CD grade 3b)	1	Reoperation
Evisceration (CD grade 3a)	1	Primary closure
Urine leakage from the bladder wall (CD grade 3a)	1	Bilateral nephrostomy catheter
Pleural effusion (CD grade 3a)	2	Percutaneous drainage catheter
Total 7(31.8%)		

**Table 5 jpm-12-01790-t005:** Histopathological findings.

Endometrial Carcinoma	7(31.8%)
Carcinosarcoma	5(22.7%)
Uterine leiomyosarcoma	5(22.7%)
Endometrial stromal sarcoma	3(13.6%)
Undifferentiated carcinoma	1(4.5%)
Serous carcinoma	1(4.5%)

**Table 6 jpm-12-01790-t006:** Disease-free survival and overall survival.

	Mean Survival Months (95% CI)	Five-Year Survivals, %
Overall survival	45.39 (34.16–56.62)	57%
Disease-free survival	36.80 (26,043–57.57)	36%

## Data Availability

The study did not report any data.

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
