# Peer review of "Is There a Role for Cytoreductive Surgery and Hyperthermic Intraperitoneal Chemotherapy in Peritoneal Carcinomatosis Due to Uterine Cancer?"

_jpm, 2022, doi:10.3390/jpm12111790_

Round 1

Reviewer 1 Report

I would like to congratulate the authors on their study.

I Would like to make the following suggestions and I would request the authors to kindly provide the following details.

1. What was the median follow up?

2. Was the HIPEC performed even for CC2/CC3?

3. The survival reported is for all the CC score or only CC0?

4. What was the interval between CRS+HIPEC intervention and the first recurrence?

5. What was the method of HIPEC used? closed or open?

6. Could you perform a univariate analyses(Log rank/ chi square) and report the impact of histology and PCI and CCscore on survival.

Author Response

Response to Reviewer 1 Comments

Thank you for your feedback. Here are the changes and responses.

  1. What was the median follow up?

Response 1: Median follow up was 32 (6-60) months. This is added to the results section

  1. Was the HIPEC performed even for CC2/CC3?

Response 2: HIPEC was not performed to CC2/CC3 cases.

  1. The survival reported is for all the CC score or only CC0?

Response 3: HIPEC is performed to CC0 and CC1 cases, therefore the survival reported is for these patients only

  1. What was the interval between CRS+HIPEC intervention and the first recurrence?

Response 4: Earliest recurrence was observed in post operative 12th month. This is also added to the results section.

  1. What was the method of HIPEC used? closed or open?

Response 5: All cases of HIPEC was performed in closed technique. This is also added to the materials and method section.

  1. Could you perform a univariate analyses(Log rank/ chi square) and report the impact of histology and PCI and CCscore on survival

Response 6: the primary aim of this study was to analyze the effect of CRS and HIPEC on uterine cancer. Performing a univariate analysis could be a topic for another study.

Reviewer 2 Report

This is a topic that is very relevant following acceptance of CRS/HIPEC for ovarian cancer. Often, it is unclear whether those results can be applied to other GYN malignancies, especially uterine cancer. There are scattered reports of efficacy of HIPEC for uterine cancer, and my institution and I perform it in select patients, though there is bias among others not to offer HIPEC for uterine carcinomatosis. This is a nice cohort of patients, and contributes to the evidence supporting HIPEC for uterine disease. It is balanced in its conclusions, and well presented.

Minor questions would include more about the methodology. Outcomes may depend not only on the type/duration of chemotherapy, but the intra-op management (e.g., PMID 30278972). Can the authors describe more about the intra-op and post-op management of their cohort?

Author Response

Response to Reviewer  2 Comments

Thank you for your feedback. Here are the changes and responses.

Minor questions would include more about the methodology. Outcomes may depend not only on the type/duration of chemotherapy, but the intra-op management (e.g., PMID 30278972). Can the authors describe more about the intra-op and post-op management of their cohort?

  • Response 1: In this patient group the main goal is to perform CC 0/1 resection. If we can’t achieve this goal, there is no point in performing HIPEC. If the surgery is performed successfully, HIPEC can have positive results. Intraoperative and post operative management (restrictive fluid resuscitation, avoiding packed blood cells, peroparative heating etc ) is effective in mortality and morbidity.In preoperative period, we use drainage catheters to drain ascites. In intraoperative period, picca catheter monitorization and monitoring urine output is used in order to avoid excessive fluid resuscitation. Heating is used to avoid hypethermia. During HIPEC heat probe placed in esophagus can help us avoid the negative effects of hyperthermia. Closed technique should be used for HIPEC so that the intraperitoneal temperature could stay stable and decreases the exposure to chemotherapeuticals and also decreases the effect on medical staff. In the postoperative period, ERAS protocoles are performed(early feeding, early mobilization). More detailed information (PMID: 31819307) can be found.

Round 2

Reviewer 1 Report

none